# The shifting landscape of private healthcare providers before and during the COVID-19 pandemic: Lessons to strengthen the private sectors engagement for future pandemic and tuberculosis care

**Rodiah Widarna**[1]*, **Nur Afifah**[1], **Hanif Ahmad Kautsar Djunaedy**[1], **Angelina Sassi**[2], **Nathaly Aguilera Vasquez**[2], **Charity Oga-Omenka**[2,3], **Argita Dyah Salindri**[1], **Bony Wiem Lestari**[1,4,5], **Madhukar Pai**[2], **Bachti Alisjahbana**[1,4,6]*

**1** Tuberculosis Working Group, Research Center for Care and Control of Infectious Diseases Universitas Padjadjaran (RC3ID Unpad), Bandung, Indonesia, **2** McGill International TB Centre, Research Institute of the McGill University Health Centre, Montreal, Quebec, Canada, **3** School of Public Health Sciences, University of Waterloo, Waterloo, Ontario, Canada, **4** Department of Public Health, Faculty of Medicine, Universitas Padjadjaran, Bandung, Indonesia, **5** Department of Internal Medicine, Radboud Institute for Health Sciences, Radboud University Medical Center, Nijmegen, The Netherlands, **6** Department of Internal Medicine, Faculty of Medicine, Hasan Sadikin General Hospital, Universitas Padjadjaran, Bandung, Indonesia

* rodiah1101990@gmail.com (RW); b.alisjahbana@unpad.ac.id (BA)

## Abstract

### Introduction

COVID-19 pandemic changed many aspects of healthcare services and deliveries, including among private healthcare providers (i.e., private healthcare facilities [HCFs] and private practitioners [PPs]). We aimed to compare the spatial distribution of private providers and describe changes in characteristics and services offered during and before the COVID-19 pandemic, and explore the tuberculosis (TB) and COVID-19-related services offered by the private sector in Bandung, Indonesia.

### Methods

A cross-sectional study with historical comparison was conducted in 36 randomly selected community health centers areas (locally referred to as *Puskesmas*) in Bandung, Indonesia, during the COVID-19 pandemic from 5th April 2021 – 27th December 2021. Data pertaining to before the COVID-19 pandemic was abstracted from a similar survey conducted in 2017 (i.e., INSTEP study). We obtained latitude and longitude coordinates of private healthcare providers and then compared the geographical spread with data collected for INSTEP study. We also compared characteristics of, and services provided by private healthcare providers interviewed during the COVID-19 pandemic with those previously interviewed for INSTEP study. Differences were summarized using descriptive and bivariate analyses.

**Data Availability Statement:** The data that support the findings of this study are openly available in openICPSR at https://doi.org/10.3886/E199007V1.

**Funding:** This work was supported, in whole or in part, by the Bill & Melinda Gates Foundation [INV-022420]. Under the grant conditions of the Foundation, a Creative Commons Attribution 4.0 Generic License has already been assigned to the Author Accepted Manuscript version that might arise from this submission.

**Competing interests:** Prof. Madhukar Pai is the Editor-in-Chief of PLOS Global Public Health

## Results

From April–December 2021, we surveyed 367 private HCFs and interviewed 637 PPs. Compared to INSTEP study data, the number of operating HCFs was reduced by 3% during the COVID-19 pandemic (401 vs. 412 before COVID-19), although we observed increases in laboratory service (37.8% increase), x-ray service (66.7% increase), and pharmacy (18.1% increase). Among a subset of private HCFs managing patients with respiratory tract infection symptoms, a quarter (60/235, 25.3%) indicated that they had to close their facilities in response to the emerging situation during the COVID-19 pandemic. For PPs, the number of practicing PPs was reduced by 7% during the COVID-19 pandemic (872 vs. 936 before COVID-19). Interestingly, the number of practicing PPs encountering patients with TB disease increased during the COVID-19 pandemic (42.9% vs. 35.7% before COVID-19, p = 0.008).

## Conclusion

This study confirmed that the COVID-19 pandemic adversely impacted health care service deliveries in private sectors, largely marked by closures and shortened business hours. However, the increased service capacities (laboratory and pharmacy), as well as significant increase in the number of patients cared for TB disease by PPs during the COVID-19 pandemic, made a more compelling case to further the implementation of public-private mix model for TB care in Indonesia.

## Introduction

The coronavirus disease 2019 (COVID-19) pandemic, declared by the World Health Organization (WHO) from Jan 2020 –May 2023 [1], impacted many aspects of human life, including social capital, economy, education system, as well as global mobility [2]. Importantly, border closures and movement restrictions implemented during the COVID-19 pandemic impacted healthcare deliveries globally. However, little is known to what extent these disruptions impacted healthcare deliveries in low- or middle-income countries where the COVID-19 burden was far higher than in high-income countries.

In Indonesia, about two-thirds of the nation's health market is dominated by the public sector, where the majority of government resources are being allocated. Private healthcare providers, accounted for the reminder one-third of Indonesia's health market, remains a vital pillar of the nation's health system and it has been grown quite significantly in the past few decades due to the increasing demands for more modern equipment and treatment, especially from individuals with high-income [3]. However, it is unknown whether private sectors in Indonesia played a key role in healthcare delivery during the COVID-19 pandemic [4, 5].

A cross-sectional survey conducted in 15 Indian states reported that 40% of private providers surveyed had to close their facilities in 2020 due to the implementation of infection control measures and movement restrictions [6]. Although most healthcare facilities did not change their service and/or costs in this Indian study [6], several others reported reduced business hours and a substantial decrease in number of patient visits [7–10]. Some studies also underscored the role of private sectors in supporting public sectors in delivering healthcare services during the COVID-19 pandemic [11, 12]. While our previous study suggested that private

sectors showed some degree of adaptability by transitioning their care to telemedicine [13] during the COVID-19 pandemic, to date, there is no evidence from the field comparing the healthcare services provided by the private sectors before and after COVID-19 pandemic.

Given this knowledge gap, we aimed to describe a) the spatial distribution and the density of private healthcare facilities (HCFs) before and during the COVID-19 pandemic, b) changes in services offered by private HCFs and private practitioners (PPs) before and during the COVID-19 pandemic, and c) TB and COVID-19 related services offered by the private sectors during the COVID-19 pandemic in Bandung, Indonesia.

We then used these informations to identify gaps and opportunities in efforts to engage the private sector in the national TB care program. Our previous work, conducted among pulmonary TB patients in Bandung, West Java, Indonesia, estimated that 75% participants first sought care and 40% started treatment in private healthcare facilities [14, 15]. Thus, understanding how private healthcare providers' landscape is changing during the COVID-19 pandemic to better understand the potential impacts of the pandemic on TB care and management are critical for more effective public-private mix (PPM) model implementation.

## Materials and methods

### Study design

The present paper is part of a parent study, the "*COVID Impact on Private Health Markets*" (COVET) study [10, 16–18], which aimed to a) investigate the impact of the COVID-19 pandemic on the functioning of private healthcare markets in three high-TB burden countries (Indonesia, India, Nigeria) and b) identify the implications of the COVID-19 pandemic on private providers engagement for TB care. Using data from Indonesia, we conducted a cross-sectional study [19] among private HCFs and PPs in Bandung, West Java, Indonesia [20], with a historical comparison to a similar survey conducted in 2017 (INSTEP study). We collected geospatial and health service information from single and multiple providers (i.e., single-provider, primary- and secondary-level HCFs) identified during the tracing and mapping process from 5th April 2021 – 24th December 2021.

### Setting

Similar to our previous study entitled "*Investigation of Health Services for TB by External Private Providers*" (2017 INSTEP) study [20], the COVET Indonesia sub-study was conducted among private healthcare facilities within the catchment area in 36 (out of 80) randomly selected community health centers (CHCs, locally referred to as *Pusat Kesehatan Masyarakat (Puskesmas)* in Bandung from April 5th–December 27th, 2021 (2021 COVET). Of note, Bandung City's healthcare infrastructure comprises hospitals (either government or privately controlled), private health clinics, and CHCs (TB care is primarily provided by CHCs). In 2021, West Java had the highest number of TB cases in Indonesia, with the TB incidence rate of 346 per 100,000 population for Bandung, the capital of West Java; this was increased to 562 per 100,000 population in 2022 [21–23]. In 2021, there were 37,917 COVID-19 cases (6.7 times higher than 2020 statistics) and 1269 COVID-19 deaths (8.2 times higher than 2020 statistics) reported in Bandung [22].

For the present study, we recruited and trained 11 enumerators from 19th February 2021 – 7th October 2021. From 5th April– 24th December 2021, enumerators explored CHC coverage areas selected in the INSTEP study to identify operating HCFs. We also used data previously collected for INSTEP study and cross-checked with HCFs data provided by CHCs to identify HCFs that may have been missed during the mapping and survey process. Geospatial information was collected using Global Positioning System (GPS) software installed in

enumerators' mobile devices. Enumerators geotagged the location upon discovery and noted if the HCF was still operating. Enumerators then recorded clinic details, including the name of the clinic, qualification of practicing private practitioners, clinic time operation, etc., in the HCFs survey questionnaire (i.e., mapping survey). Permanently closed HCFs were noted in the same questionnaire. All the data from the application were linked to a web-based database using Research Electronic Data Capture (REDCap) electronic data capture tools hosted at Universitas Padjadjaran. REDCap is a secure, web-based software platform that supports data capture for research studies [24]. From 15th May 2021 – 27th December 2021, after geotagging and collecting general information for the clinics, enumerators surveyed private practitioner(s) (PPs) at the facility with a PP questionnaire (i.e., provider survey). General practitioners (GPs) and specialists were interviewed to collect information on demographic characteristics, qualifications, experiences in encountering TB patients, and changes in patient trends and clinic operations (i.e., including service charges, types of services offered) in response to the COVID-19 pandemic. This information was recorded in the provider survey, also collected using the mobile version of REDCap.

## Participants

We used convenience sampling method to select individuals working in private healthcare facilities to be surveyed in the present study. Clinic managers, clinic owners, doctor(s) in charge, general practitioners, or other clinic staff were eligible to be interviewed for the mapping survey. For the providers survey, eligible study participants included GPs and specialists.

## Definitions

We defined "before COVID-19" as the same period of INSTEP study (i.e., August 2017 –April 2018), whereas "during COVID-19" was defined as the middle phase of COVID-19 pandemic in Indonesia (April 2021 –December 2021). Our primary study outcomes included a) changes in the spatial pattern of private HCFs, b) changes in characteristics of and services provided by private HCFs and practitioners before and during the COVID-19 pandemic, and c) TB- and COVID-19 related services offered by private HCFs during the COVID-19 pandemic.

We calculated HCFs densities by dividing the number of HCFs found during the mapping period by the actual population size corresponding to CHC coverage areas in each study phase and were expressed per 100,000 population. The 2021 TB notification rates were obtained from the Bandung Municipal Health Department and were also expressed per 100,000 population. We grouped different types of HCFs according to their service level, and classified them into a) single provider HCFs (i.e., healthcare/services provided by a single GP), b) primary-level HCFs (i.e., healthcare/services provided by at least two GPs), and c) secondary-level HCFs (i.e., healthcare/services provided by at least one specialist and other specialist[s]/GPs). We classified PPs according to their qualifications and categorized them into a) GPs (i.e., doctors who qualified in general medical practice) and b) specialists (i.e., doctors who completed a post-graduate residency program in a specific medical field). We also categorized PPs according to whether or not they are managing patients with respiratory tract infection (RTI) symptoms, such as cough, fever, runny nose, and/or dyspnea.

## Statistical methods

Locations of healthcare facilities were pinpointed and mapped with the Quantum Geographic Information System (QGIS) application [25]. Descriptive analyses were used to summarize characteristics of private healthcare facilities and providers surveyed for the present study. Bivariate analyses (i.e., chi-square test for categorical variables and Wilcoxon rank sum tests

for continuous variables) were used to compare characteristics of, as well as services offered by private healthcare facilities and providers before and during the COVID-19 pandemic. Measures of association were expressed as proportion differences (PD) and 95% confidence intervals (CI). All analyses were performed in IBM SPSS Statistics for Windows, version 25 (IBM Corp., Armonk, N.Y., USA), with p≤0.05 considered significant in all analyses.

## Ethics

This study was approved by the Research Ethics Committee of Universitas Padjadjaran, Bandung, Indonesia (No.166/UN6.KEP/EC/2021), Bandung Municipal Health Office (No. PP.06.02/5603/Dinkes/II.2021 and No.PP.06.02/157.63/Dinkes/X/2021), Bandung National Unity and Politics Agency (No.PP.09.01/410-kesbangpol/IV/2021 and No.PP.09.01/1549-kesbangpol/X/2021), and the Institutional Review Board (IRB) at the McGill University Health Centre (Covid BMGF/2021-7197). All survey participants provided written informed consent before study procedures.

## Results

During the COVET study period, our enumerators identified a total of 641 HCFs from the 36 selected CHC areas (Fig 1). Of these, 61.5% (394/641) were identified from INSTEP study (i.e., before COVID-19), and 38.4% (246/641) were newly identified. There were 132 HCFs identified from INSTEP study and 108 newly identified HCFs (total 240/641, 37.4%) that were not operating at the time of mapping (i.e., HCFs were either temporarily or permanently closed). The majority (367/401, 91.5%) of operating HCFs were successfully interviewed by our

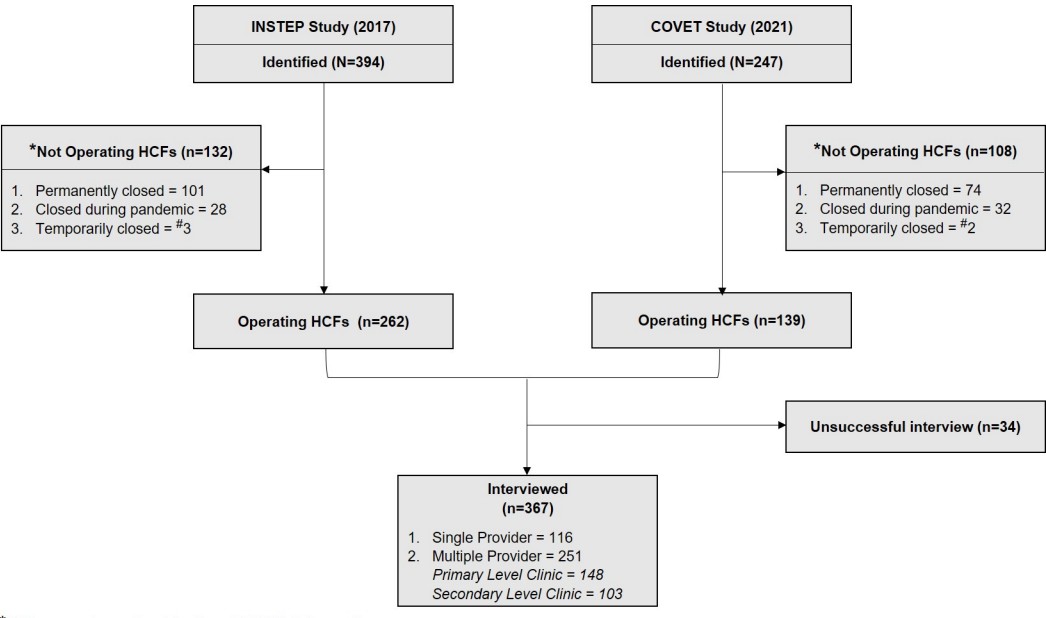

*HCFs were not operating at the time of COVET study mapping
#Some of the reasons of closures included HCFs were in process of extending the practice permit, doctors were continuing study, or HCFs were being renovated.

**Fig 1. Flowchart depicting the flow and inclusion of private healthcare facilities mapped and interviewed in Bandung during COVET study period (5th Apr 2021– 24th Dec 2021).** We identified 394 HCFs from our prior study (INSTEP study) conducted in 2017; of these, 132 were not operating during the COVET study mapping process. There were 247 HCFs that we newly identified during COVET study (and these were not identified during INSTEP study), but 108 were not operating at the time of COVET study mapping process. Of 401 operating HCFs, we successfully interviewed 367, which were included in our primary analyses.

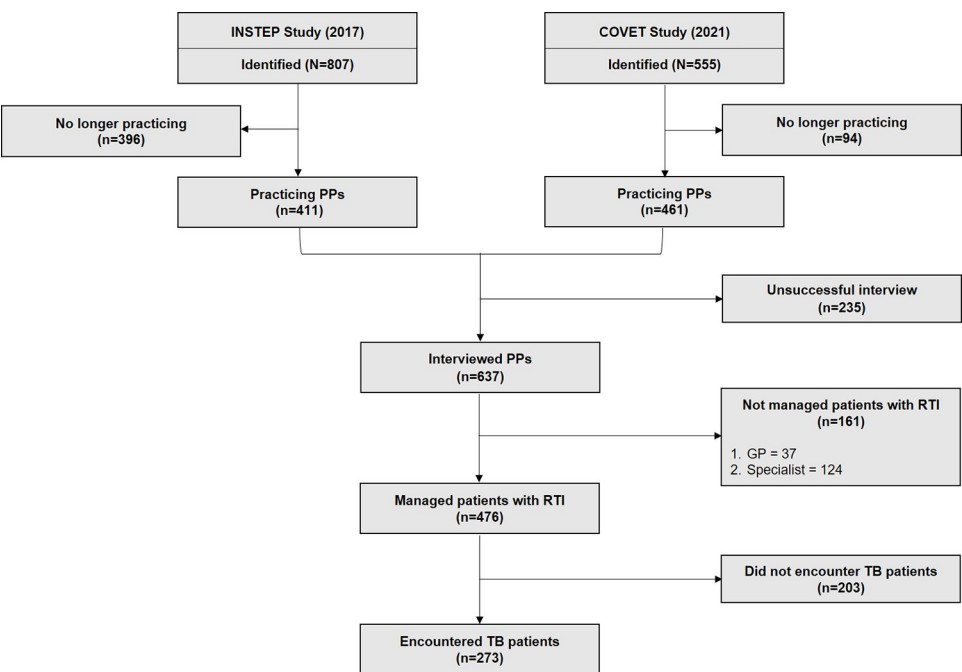

**Fig 2. Flowchart depicting the flow and inclusion of private practitioners mapped and interviewed in Bandung during COVET studies period (15th May 2021 – 27th Dec).** We identified 807 PPs from our prior study (INSTEP study) conducted in 2017; of these, 396 were not operating during the COVET study mapping process. There were 555 PPs that we newly identified during COVET study (and these were not identified during INSTEP study), but 94 were not operating at the time of COVET study mapping process. Of 872 operating PPs, we successfully interviewed 637, which were included in our primary analyses.

enumerators, with two-thirds (251/367, 68.4%) classified as either primary or secondary level clinics (i.e., HCFs with multiple providers).

Our enumerators identified a total of 1362 PPs from the 36 selected CHC areas (Fig 2). Of these, 59.3% (807/1362) were identified from INSTEP study, and 40.7% remaining (555/1362) were newly identified (i.e., currently registered in the local health office database but were not identified during INSTEP study). There were 396 PPs identified from INSTEP study and 94 newly identified PPs (total 490/1362, 36.0%) that were no longer practicing at the time of mapping due to either doctor-related (e.g., doctor resigned, deceased, retired, or continuing study) or HCF-related reasons (e.g., HCFs permanently closed, closed during pandemic, or temporarily closed). Of 872 practicing PPs at the time of mapping, 73% (637/872) were successfully interviewed, and 74.7% (476/637) of which self-reported that they are managing patients with RTI symptoms. Of these, a subset (273/476, 57.4%) also reported that they are encountering patients with TB disease.

## Spatial distribution of private healthcare providers before and during the COVID-19 pandemic

We observed no significant changes regarding the spread of private HCFs before and during the COVID-19 pandemic (Fig 3A and 3B). Among the 36 randomly selected CHC areas, the median density of HCFs per 100,000 population before COVID-19 was 29 (interquartile range [IQR] 17–41) vs. 28 (IQR 17–37) during COVID-19 (p = 0.778). Importantly, in the map where we contrasted HCF density and TB notification rate before and during the COVID-19 pandemic, we observed that areas with lower HCF density also had higher TB notification rates.

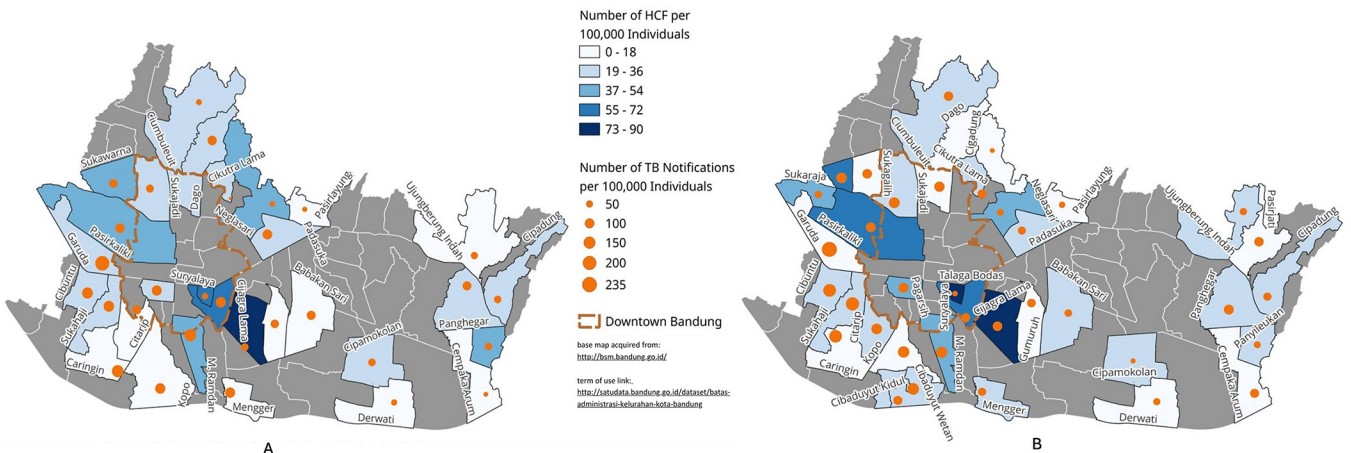

**Fig 3.** Maps of private healthcare facilities' density and TB Notifications rates relative to population size in included study areas during INSTEP (A) and COVET (B) study periods, Bandung [26]. The same area as in the INSTEP study, but has experienced an expansion in the number of health centers compared to the INSTEP study. There were 36 study areas included in COVET study. The blue color gradation shows the density of healthcare facilities relative to the population size; darker shades of blue indicate areas with higher HCFs density (i.e., the area has more operating healthcare facilities per 100,000 population). The orange dots with differing sizes indicate the magnitude of TB notification rates in the study areas. Orange dashed-line marks the downtown Bandung area, which is more crowded compared to other study areas. Printed texts are the names of selected Community Health Centers (CHCs) study areas.

## Characteristics of private healthcare providers before and during the COVID-19 pandemic

Overall, the number of HCFs operating during the COVID-19 pandemic was slightly decreased compared to the number before the COVID-19 pandemic (401 vs. 412, ~2.7% relative decrease). Of HCFs that were successfully interviewed, the proportion of HCFs managing patients with RTI symptoms during the COVID-19 pandemic was reduced by 12.0% (308 vs. 350 before COVID-19) (**Fig 4**). However, the number of HCFs with diagnosis-related services such as laboratory and X-ray increased during the COVID-19 pandemic (37.8% increase for laboratory services and 66.7% increase for x-ray services). Additionally, the number of HCFs with pharmacy services also substantially increased by 18.1% during the COVID-19 pandemic. Furthermore, the number of HCFs with established network to the national universal health insurance (locally known as *BPJS Kesehatan*) also increased during the COVID-19 pandemic (36.1% increased). The breakdown of absolute changes in healthcare services according to HCF types are depicted in **S1 Fig**.

Overall, the number of practicing PPs during the COVID-19 pandemic was significantly decreased compared to the number before the COVID-19 pandemic (872 vs. 936, relative decrease (7%), $p < 0.001$) [20]. The demographic characteristics and qualifications of PPs interviewed during the COVID-19 pandemic were similar to those interviewed before the COVID-19 pandemic (**Table 1**). Among PPs who were successfully interviewed, the proportion of PPs managing patients with RTI symptoms was significantly decreased during COVID-19 (476/637, 74.7%) compared to the period before the COVID-19 pandemic (594/674, 88.1%) (relative change -19.9% ((476–594)/594*100%), $p = <0.001$). Notably, the proportion of PPs encountering TB patients was substantially increased during the COVID-19 pandemic compared to the proportion before the COVID-19 pandemic (42.9% vs. 35.7%, ~13.3% ((273–241)/241*100%) increase, $p = 0.008$).

Among PPs we successfully interviewed, the proportion of general practitioners were slightly lower during the COVID-19 pandemic (449/637, 70.5%) when compared to the proportion before the COVID-19 pandemic (494/674, 73.3%) (**Table 1**). The proportions of

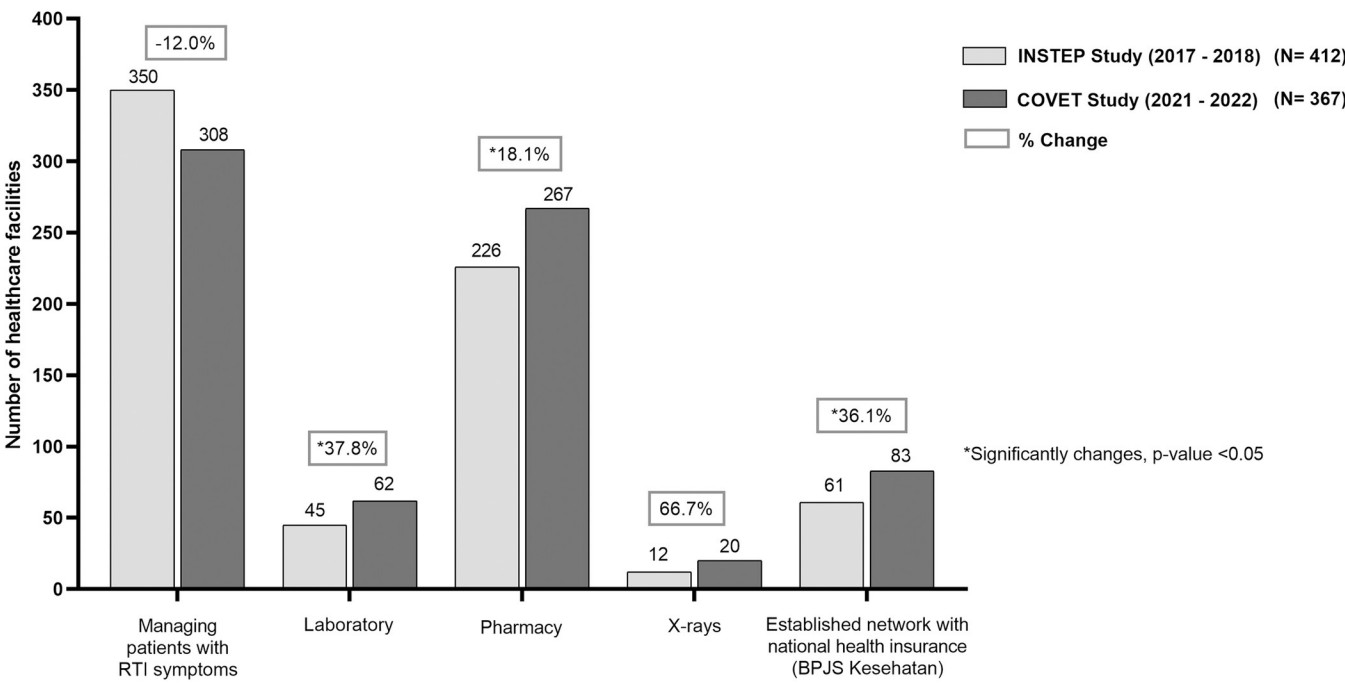

**Fig 4. Changes in services provided by private healthcare facilities interviewed during INSTEP and COVET studies.** Percent changes (presented in %) were calculated by subtracting the absolute numbers observed during COVET and INSTEP study then divided by the number observed during INSTEP study.

successfully interviewed pulmonologists and internists increased during COVID-19 (100% increase for pulmonologists and 40.9% increase for internists). The trend was consistent even when we restricted our analyses to PPs encountering patients with TB (**S2 Fig**). Among PPs who were successfully interviewed during the two study periods, the majority reported that their clinic opens weekdays only, although we observed higher proportion PPs practicing during both weekdays and weekend during COVET study period (38.8% vs. 28.3% during INSTEP study period, p<0.001) (**Table 1**). A similar trend was observed when we restricted the analyses to PPs managing patients with RTI symptoms (**S1 Table**).

## Impact of COVID-19 and TB services offered by private healthcare providers during the COVID-19 pandemic

Among HCFs managing patients with RTI symptoms, approximately a quarter (60/235, 25.5%) had to temporarily close their facilities at some point during the COVID-19 pandemic, with the highest percentage of closures observed among single providers (42.3% vs. 18.8% among secondary-level HCFs and 16.5% among primary-level HCFs, p<0.001) (**Table 2**). Changes in HCFs business hours during the COVID-19 pandemic were common and most frequently reported among primary-level HCFs (56.9% vs. 42.3% among single providers HCFs and 37.5% among secondary-level HCFs, p = 0.037). Among HCFs that changed their business hours, the majority (>90%) reported that they had to shorten their business hours during the COVID-19 pandemic. Other characteristics (e.g., detailed of discontinued services during the COVID-19 pandemic, number of visiting patients per day, etc.) are described in **S2 Table**.

**Table 1. Characteristics of private practitioner interviewed in INSTEP and COVET studies.**

| Characteristics | INSTEP Study | COVET Study | Relative change[†] | p-value[‡] |
|---|---|---|---|---|
|  | (n = 674) | (n = 637) |  |  |
|  | n (%) | n (%) |  |  |
| Age, years, *median (IQR)* | 40 (30–53) | 39 (31–51) | - | 0.758 |
| *Missing* | 137 | 26 |  |  |
| Male | 305 (45.3) | 257 (40.3) | -15.7 | 0.073 |
| Managed patients with RTI symptoms | 594 (88.1) | 476 (74.7) | -19.9 | **<0.001** |
| Encountered TB patients | 241 (35.7) | 273 (42.9) | 13.3 | **0.008** |
| **Qualification** |  |  |  |  |
| General Practitioner | 494 (73.3) | 449(70.5) | -9.1 | 0.201 |
| Specialist | 180 (26.7) | 188 (29.5) | 4.4 |  |
| *Pulmonologist* | *0/180 (0.0)* | *7/188 (3.8)* | 100 | N/A[#] |
| *Pediatrician* | *26/180 (14.4)* | *20/188 (10.6)* | -23.1 |  |
| *Internist* | *22/180 (12.2)* | *31/188 (16.5)* | 40.9 |  |
| **Number of doctors practicing in HCFs** |  |  |  |  |
| Single provider HCF* | 144 (21.4) | 100 (15.7) | -30.6 | **0.008** |
| Multiple provider HCF | 530 (78.6) | 537 (84.3) | 1.3 |  |
| *Primary level HCF*** | *307/530 (57.9)* | *291/537 (54.2)* | -5.2 | 0.220 |
| *Secondary level HCF**** | *223/530 (42.1)* | *246/537 (45.8)* | 10.3 |  |
| **Schedule of Practice** |  |  |  |  |
| Weekdays only | 436 (64.7) | 313 (49.1) | -28.2 | **<0.001** |
| Weekend only | 24 (3.6) | 22 (3.5) | -8 |  |
| Both weekdays and weekend | 191 (28.3) | 247 (38.8) | 29 |  |
| By appointment only | 19 (2.8) | 55 (8.6) | 189 |  |
| *Missing* | 4 | 0 |  |  |
| Time of practice, total days per week, *median (IQR)* | 4 (3–5) | 3 (2–5) | - |  |

*Healthcare facility (HCF) with healthcare/services provided by a single general practitioner

**Healthcare facility (HCF) with healthcare/services provided by at least two general practitioners

***Healthcare facility (HCF) with healthcare/services provided by at least one specialist and other specialist(s)/general practitioner(s)

[†]Relative change (presented in percent, %) was calculated using this formula: (COVET–INSTEP)/INSTEP

[‡]p-values were calculated using chi-square tests unless indicated otherwise; missing values were excluded from chi-square tests but presented in the table

[#]We did not show the p-value here since there were several operating pulmonologists but no pulmonologist was successfully interviewed during INSTEP study; this prevented us from making any statistical inference.

Abbreviations

COVET–COVID Impact on Private Health Markets; INSTEP–Investigation of Health Services for TB by External Private Providers; HCF–Healthcare facility; IQR–Interquartile Range (25–75); RTI–Respiratory Tract Infection; TB–Tuberculosis

**Bold** indicates that the finding is statistically significant with a≤0.05

About two-thirds of primary- and secondary-level HCFs (64.8% for primary-level HCFs and 62.5% for secondary-level HCFs) offered COVID-19-related services, compared to 26.9% of single-provider HCFs (p<0.001) (**Table 2**). Of HCFs offering COVID-19-related services, the majority (73/121, 60.3%) offered only COVID-19 testing and did not treat confirmed patients in their facilities. Most of HCFs required staff to wear PPE during the COVID-19 pandemic (71.8% among single provider HCFs; 83.5% among primary-level HCFs; and 93.8% among secondary-level HCFs). Other challenges HCFs had to face during the COVID-19 pandemic included financial hardship leading to staff lay-offs (4.3%), staffing shortages (19.6%), and shortages of laboratory consumables and medications (3.4% and 17.4%, respectively), with the majority were reported among primary-level HCFs.

**Table 2. Impact of COVID-19 pandemic on services deliveries stratified by types of private healthcare facilities during COVET study period (N = 235)§.**

| Characteristics | Single provider HCF* | Primary level HCF** | Secondary level HCF*** | p-value† |
|---|---|---|---|---|
| | (n = 78) | (n = 109) | (n = 48) | |
| | n (%) | n (%) | n (%) | |
| Facilities were ever closed due to COVID-19 pandemic | 33 (42.3) | 18 (16.5) | 9 (18.8) | <**0.001** |
| **Changes in operating hours for facility** | | | | |
| No | 45 (57.7) | 47 (43.1) | 30 (62.5) | **0.037** |
| Yes | 33 (42.3) | 62 (56.9) | 18 (37.5) | |
| *Longer hours* | *0/33 (0.0)* | *2/62 (3.2)* | *0/18 (0.0)* | |
| *Shorter hours* | *30/33 (91.0)* | *58/62 (93.6)* | *18/18 (100.0)* | |
| *Others‡* | *3/33 (9.0)* | *2/62 (3.2)* | *0/18 (0.0)* | |
| **Changes in healthcare services offered pre- and during COVID-19 pandemic** | | | | |
| Lab services | 2 (2.6) | 6 (5.5) | 0 (0.0) | 0.190 |
| Pharmacy services | 2 (2.6) | 2 (1.8) | 0 (0.0) | 0.552 |
| TB screening | 0 (0.0) | 1 (0.9) | 0 (0.0) | 0.560 |
| TB testing | 0 (0.0) | 1 (0.9) | 0 (0.0) | 0.560 |
| **Facility offered any COVID-19 related services** | | | | |
| No | 57 (73.1) | 38 (35.2) | 18 (37.5) | <**0.001** |
| Yes | 21 (26.9) | 70 (64.8) | 30 (62.5) | |
| *COVID-19 testing only* | *12/21 (57.1)* | *44/70 (62.9)* | *17/30 (56.7)* | |
| *COVID-19 treatment only* | *6/21 (28.6)* | *1/70 (1.4)* | *4/30 (13.3)* | |
| *COVID-19 testing and treatment* | *3/21 (14.3)* | *25/70 (35.7)* | *9/30 (30.0)* | |
| *Missing* | 0 | 1 | 0 | |
| **Adjustments made as part of infection control measures during COVID-19 pandemic** | | | | |
| Limiting the number of clients seen per day | 27 (34.6) | 31 (28.4) | 15 (31.3) | 0.667 |
| Limiting the number of clients in the building at a given time | 29 (37.2) | 53 (48.6) | 21 (43.8) | 0.298 |
| Limiting the number of health officer/staff in the building at a given time | 4 (5.1) | 13 (11.9) | 4 (8.3) | 0.271 |
| Mandating the use of PPE for all health officer and/or staff | 56 (71.8) | 91 (83.5) | 45 (93.8) | **0.007** |
| Mandating the use of PPE for all patients | 11 (14.1) | 22 (20.2) | 8 (16.7) | 0.551 |
| Increasing service fee | 0 (0.0) | 3 (2.8) | 1 (2.1) | 0.348 |
| Others# | 9 (11.5) | 13 (11.9) | 1 (2.1) | 0.131 |
| **Administrative changes in the facility due to COVID-19 pandemic** | | | | |
| Needed to lay off any staff due to financial reason | 3 (3.8) | 7 (6.4) | 0 (0.0) | **0.008** |
| Experienced staffing shortage during pandemic | 7 (9.0) | 31 (28.4) | 8 (16.7) | **0.001** |
| Experienced PPE shortage in the last month | 6 (7.7) | 11 (10.1) | 4 (8.3) | 0.825 |
| Experienced testing reagents shortage in the last month | 1 (1.3) | 7 (6.4) | 0 (0.0) | **0.010** |
| Experienced medications shortage in the last month | 4 (5.1) | 30 (27.5) | 7 (14.6) | **0.005** |
| **Disruption due to COVID-19 pandemic** | | | | |
| Short term disruption (<3 months) in medical supplies | 23 (29.5) | 42 (38.5) | 18 (37.5) | 0.416 |
| Long term disruption (>3 months) in medical supplies | 12 (15.4) | 27 (24.8) | 9 (18.8) | 0.277 |
| Short term disruption (<3 months) in staff shortages | 2 (2.6) | 9 (8.3) | 6 (12.5) | 0.096 |
| Long term disruption (>3 months) in staff shortages | 2 (2.6) | 7 (6.4) | 0 (0.0) | 0.120 |
| Renovation for infection control measures | 6 (7.7) | 15 (13.8) | 5 (10.4) | 0.422 |

*(Continued)*

**Table 2.** (Continued)

| | | | | |
|---|---|---|---|---|
| Others<sup>‖</sup> | 37 (47.4) | 27 (24.8) | 16 (33.3) | **0.005** |

*Healthcare facility (HCF) with healthcare/services provided by a single general practitioner

**Healthcare facility (HCF) with healthcare/services provided by at least two general practitioners

***Healthcare facility (HCF) with healthcare/services provided by at least one specialist and other specialist(s)/general practitioner(s)

†p-values were calculated using chi-square tests unless indicated otherwise

‡Other changes in operating hours included by appointment, open with less practice days, changing time of opening and closing hours

#Other adjustment included all covid health protocols, by appointment, changing operating hours, cheaper cost, fewer operating hours, just open for some patients, opening hours change, reject covid patient, room disinfection, room sterilization, services restriction, telemedicine, just for health national

‖Other disruptions due to COVID-19 pandemic included changes in health protocol and limiting number of patients

§Data were obtained from interviews with doctors managing patients with RTI symptoms, representing each healthcare facilities included in COVET study

Abbreviations

COVET–COVID Impact on Private Health Markets; HCF–Healthcare facility; PPE–Personal Protective Equipment

**Bold** indicates that the finding is statistically significant with a≤0.05

The COVID-19 pandemic resulted in similar changes to practices at the PP level (**Table 3**). Private specialists were more likely to isolate and treat patients with COVID-19 in their facility (20.3%) compared to private GPs (6.8%) (p = 0.001). Although non-significant, private specialists were more likely to test for both TB and COVID-19 when encountering patients with RTI symptoms compared to private GPs (28.1% vs. 23.5%, p = 0.427).

We also describe TB related services among private practitioners in **Table 4**. When stratified according to PPs' qualifications, GPs diagnosed more TB patients (64.8%, 147/227) than specialists (19.6%, 9/46, p<0.001), although specialists were more likely to offer treatment after diagnosing TB cases (67.4%, 31/46) compared to GPs (25.5%, 58/227, p<0.001). Furthermore, pulmonologists reported the highest number of TB diagnoses in the past month (median = 20, IQR 10–50) (**Fig 5A**). A similar trend was observed for the number of individuals with TB being treated by PPs with the highest median reported among pulmonologists (median = 50, IQR 10–100) (**Fig 5B**). The majority (42.7%) of PPs used single-drug formulations [27] when treating individuals with TB disease. Specialists were also more likely to treat persons with TB using a combination of both Fixed Dose Combination (FDC) and single-drug formulations, when compared to GPs (29.7% vs. 13.8%, p = 0.040) (**Table 4**). In terms of TB drugs source, both GPs (48/80, 60.0%) and specialists (29/37, 78.4%) provided TB medications largely from their own pharmacy service.

## Discussion

In our private healthcare facilities and providers survey conducted during the COVID-pandemic, we did not see any substantial changes in the geographical spread as well as service offered by both HCFs and PPs when compared to period before the COVID-19 pandemic. However, our study findings on the increasing service capacities underscore the importance of mapping and survey among private healthcare providers especially during a situation like the COVID-19 pandemic as these may help verify the currently operating facilities as well as newly added healthcare services that may benefit TB care and control program post-pandemic.

We did not observe significant changes in the geographical distribution of HCFs before and during the COVID-19 pandemic in Bandung, West Java, Indonesia. Despite the notable growth of private sectors between the two study periods, more than one-third of the private healthcare providers (both HCFs and PPs) in our final databases were not operating at the

**Table 3. COVID-19 related services offered by private practitioners who managed RTI symptoms included in COVET study, Bandung (N = 476).**

| Characteristics | General practitioners | Specialists | Total | p-value* |
|---|---|---|---|---|
| | (n = 412) | (n = 64) | (N = 476) | |
| | n (%) | n (%) | n (%) | |
| **Testing for COVID-19 among all patients with COVID-19-like symptoms** | 248 (60.2) | 38 (59.4) | 286 (60.1) | 0.896 |
| **Testing for both COVID-19 and TB among all patients with appropriate symptoms** | 97 (23.5) | 18 (28.1) | 115 (24.2) | 0.427 |
| **Self-rated confidence to differentiate COVID and TB** | | | | |
| Very Confident | 31 (7.5) | 6 (9.4) | 37 (7.8) | 0.197 |
| Confident | 301 (73.1) | 47 (73.4) | 348 (73.1) | |
| Somewhat Confident | 68 (16.5) | 7 (10.9) | 75 (15.8) | |
| Not Confident | 10 (2.4) | 2 (3.1) | 12 (2.5) | |
| Unsure | 2 (0.5) | 2 (3.1) | 4 (0.8) | |
| **Management and referral for suspected COVID-19 patients** | | | | |
| **Testing** | | | | |
| Testing in-house | | | | |
| *Performed COVID-19 testing in the facility itself* | 210 (51.0) | 29 (45.3) | 239 (50.2) | 0.404 |
| Testing refers to other facilities | | | | |
| *Referred to different private facilities for testing* | 134 (32.5) | 34 (53.1) | 168 (35.3) | **0.001** |
| *Referred to public facility for testing* | 217 (52.7) | 23 (35.9) | 240 (50.4) | **0.013** |
| **Patient Management** | | | | |
| *Isolated patients in the facility itself* | 28 (6.8) | 13 (20.3) | 41 (8.6) | **0.001** |
| *Instructed patient to return home and self-isolate (if symptoms were mild)* | 150 (36.4) | 25 (39.1) | 175 (36.8) | 0.679 |
| *Others*** | 33 (8.0) | 5 (7.8) | 38 (8.0) | 0.996 |
| **Coordination** | | | | |
| *Contacted local or national health authorities for guidance* | 23 (5.6) | 3 (4.7) | 26 (5.5) | 0.821 |
| *Others**** | 15 (3.6) | 1 (1.6) | 16 (3.4) | 0.436 |

*p-values were calculated using chi-square tests unless indicated otherwise

**Others included provided consultation with specialist, treated patient in-house, gave vitamin or prescription, and refered patient if symptoms were severe

***Others included conducted contact tracing, educated patients, encouraged patients to report to CHC (Community Health Center) and/or to local authorities, encouraged HCF to report to CHC

Abbreviation: TB–tuberculosis

time of mapping. This finding is not surprising as the increased closure rates of hospitals and other types of health clinics during the COVID-19 pandemic has been documented globally [28, 29]. Our finding which suggests that areas with lower HCFs density were more likely to have higher TB notification rates, is consistent with our previous survey conducted before the COVID-19 pandemic [20]. This finding is critical as 73–75% of patients with TB in Indonesia may seek initial care among private sectors [14, 15, 30] and such information could be used to inform public health officials to focus and direct their resources to increase TB notifications in areas with lower HCFs density and to boost TB treatment initiation and completion rates, especially during a pandemic like COVID-19.

Despite challenges during the COVID-19 pandemic, we observed increased capacities among private healthcare providers when compared to results of our 2017 survey. These were marked by the increased proportions of diagnosis-related (e.g., laboratory and radiology services) and other services (e.g., pharmacy and linkage to universal health insurance) among HCFs. Some studies from high TB-burden countries have reported similar increased capacities for TB care among private providers during the pandemic [31–34]. Furthermore, at PPs level, we also observed that the number of private pulmonologists and internists increased during

 The shifting landscape of private providers before and during the COVID-19 pandemic

**Table 4. TB related services offered by private practitioners who managed RTI symptoms included in COVET study (N = 476).**

| Characteristics | Types of Private Practitioners | | Total | p-value* |
|---|---|---|---|---|
| | General practitioners | Specialists | | |
| | (n = 412) | (n = 64) | (N = 476) | |
| | n (%) | n (%) | n (%) | |
| **Encountered TB patients** | 227 (55.1) | 46 (71.9) | 273 (57.4) | **0.012** |
| Diagnosed TB patients only | 147/227 (64.8) | 9/46 (19.6) | 156/273 (57.1) | **<0.001** |
| Treated TB patients only | 22/227 (9.7) | 6/46 (13.0) | 28/273 (10.3) | 0.226 |
| Both (Diagnosed and treated TB patients) | 58/227 (25.5) | 31/46 (67.4) | 89/273 (32.6) | **<0.001** |
| **Diagnosed TB patients in the past months** | 205 (50.0) | 40 (62.5) | 245 (51.5) | |
| Confirmed TB diagnosis with lab examinations only (i.e., smear) | 20/205 (9.8) | 0/40 (0.0) | 20/245 (8.2) | **0.072** |
| Confirmed TB diagnosis with chest X-Rays only | 30/205 (14.6) | 2/40 (5.0) | 32/245 (13.1) | 0.217 |
| Confirmed TB diagnosis with both lab and Chest X-Rays | 141/205 (68.8) | 38/40 (95.0) | 179/245 (73.1) | **<0.001** |
| None of them | 14/205 (6.8) | 0/40 (0.0) | 14/245(5.7) | 0.134 |
| **Cumulative number of patients diagnosed per month** | 674 | 326 | 1000 | |
| *Median (IQR)* | 2 (1–4) | 3 (1–5) | 2 (1–4) | **0.019** |
| **Cumulative number of patients referred to the lab for diagnosis confirmation per month** | 487 | 312 | 799 | |
| *Median (IQR)* | 1 (1–3) | 2 (1–5) | 2 (1–3) | **0.002** |
| **Cumulative number of patients referred to undergo x-rays procedure to confirm TB diagnosis per month** | 532 | 322 | 854 | |
| *Median (IQR)* | 2 (1–3) | 3 (1–5) | 2 (1–3) | **<0.001** |
| **Ever treating TB patients** | 80 (19.4) | 37 (57.8) | 117 (24.6) | |
| **Cumulative number of TB patients treated per month (in the last 6 months)** | 284 | 533 | 817 | |
| *Median (IQR)* | 2 (0–4) | 4 (1–10) | 2 (1–5) | **0.005** |
| **Type of Drugs** | | | | |
| FDC only | 23/80 (28.7) | 10/37 (27.0) | 33/117 (28.2) | 0.847 |
| Single-drug formulations only | 37/80 (46.3) | 13/37 (35.1) | 50/117 (42.7) | 0.258 |
| Both FDC and single-drug formulations | 11/80 (13.8) | 11/37 (29.7) | 22/117 (18.8) | **0.040** |
| Other | 9/80(11.3) | 3/37 (8.1) | 12/117 (10.3) | 0.218 |
| **TB Drugs Source** | | | | |
| HCF's pharmacy | 48/80 (60.0) | 29/37 (78.4) | 77/117 (65.8) | **0.051** |
| Prescribed to other facilities | 26/80 (32.5) | 13/37 (35.1) | 39/117 (33.3) | 0.779 |
| Public Health Office (PHO/Dinkes) | 2/80 (2.5) | 0/37 (0.0) | 2/117 (1.7) | 0.332 |
| Community Health Centre (CHC/Puskesmas) | 28/80 (35.0) | 6/37 (16.2) | 34/117 (29.1) | **0.037** |
| Other | 2/80 (2.5) | 0/37 (0.0) | 2/117 (1.7) | 0.332 |

*p-values were calculated using chi-square tests unless indicated otherwise

Abbreviations

FDC–Fixed Dose Combination; HCF–Healthcare Facility; RTI–Respiratory Tract Infection; IQR–Interquartile Range (25–75); TB–tuberculosis

**Bold** indicates that the finding is statistically significant with a≤0.0

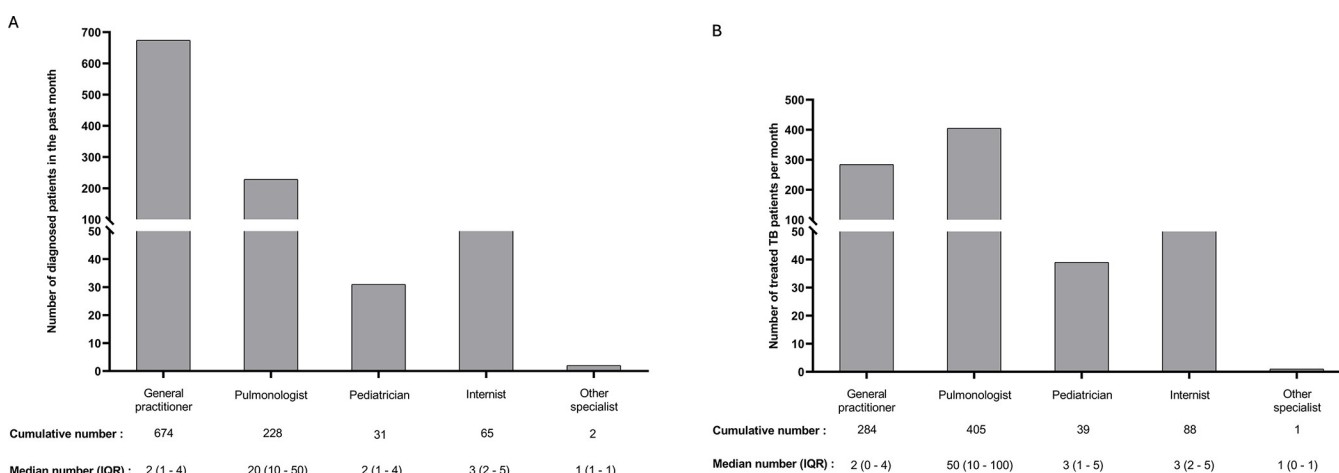

**Fig 5.** Absolute numbers and median of patients diagnosed with (A) and treated for (B) tuberculosis by private practitioners in the past month during COVET study.

the COVID-19 pandemic. Thus, identifying these increased capacities is critical as we could target specific types of HCFs or PPs to strategically engage private healthcare providers within the national TB control program and maximize efforts to identify undiagnosed or unreported TB patients in the COVID-19 era.

While COVID-19 negatively impacted healthcare service delivery among all types of private healthcare providers, the severity of the impact was different across various levels or types of HCFs. In our study, single-provider HCFs were more likely to be affected the most by the COVID-19 pandemic, as shown by the higher proportion of facility closures. This finding is expected as larger facilities (i.e., primary- and secondary-level HCFs) operated by multiple practitioners could potentially offer more flexibility in terms of business hours, types of appointments (in person vs. online) [35], and services offered during the COVID-19 pandemic. Although a cross-sectional study conducted in India suggested that many private doctors transitioned to e-health (i.e., telemedicine, social media) or offering services by appointment only due to the prevention of transmission during the COVID-19 pandemic [36], offering such flexibility is probably more challenging among single provider HCFs in Indonesia since some may open after regular business hours [20].

In terms of COVID-19-related services, the majority of private healthcare providers in Bandung offered COVID-19 testing services, but most either instructed individuals to self-quarantine or referred patients to other facilities for treatment. This is also expected as Indonesia's government directed the majority of resources to provide COVID-19-related services to public sectors, like *Puskesmas* [37–39].

Collectively, our study findings underscored several considerations for future outbreak preparedness efforts. First, engaging private sectors in an outbreak response will likely accelerate the diagnosis rate, which could suppress the disease's transmission rate. Second, different types of private HCFs showed varying degree of resilience and adaptability (including improvement in service and flexibility of healthcare deliveries via telemedicine) when faced with such challenging situation during the COVID-19 pandemic; identifying and turning these HCFs into extended government-supported clinics (e.g., fever clinic) will not only alleviate the burden of public healthcare system in delivering healthcare services but could also be a path to bringing care closer to the community. Lastly, by engaging private healthcare providers in pandemic responses, we could potentially avoid detrimental effects (e.g., psychological

distress, physical exhaustion, and high rates of COVID-19 morbidity and mortality) among healthcare workers both in public and private sectors.

## Changing landscape in private sectors: The case for strengthening the TB public-private mix model in Indonesia

Our study reported an increase in the proportion of PPs encountering individuals with TB patients during the COVID-19 pandemic. Further, pulmonologists reported a large median of both TB diagnoses as well as number of individuals being treated for TB disease in the past month at their facilities, highlighting their potential role in early identification as well as care and TB management. We further observed that medications used to treat individuals with TB disease were often obtained from the in-house pharmacy services. While this may indicate self-sufficiency, there might be a concern about adherence to the national TB treatment guidelines when the in-house pharmacy experiences medication shortages, especially during a pandemic. Thus, establishing more tailored medication supplies and treatment strategies by linking these private providers to the main executor of the national TB program (i.e., public sectors such as *Puskesmas*) are still warranted.

Our survey highlighted the potentials of incorporating private sectors in the national TB program. For instance, in the COVET study, we identified many practicing PPs who were encountering TB patients, which is likely to be significantly higher than the number of dedicated TB physicians/nurses in community health centers. Our study findings also suggested that private sectors are constantly growing, and we observed increases in capacities that could support TB diagnostic (i.e., laboratory and radiology services) and other services (i.e., pharmacy and linkage to the universal health insurance) compared to our 2017 survey. With these considerations, by incorporating private sectors in the national TB program, we could potentially reduce the TB burden through several mechanisms: a) providing wider TB catchment spots and ultimately increasing TB notification rates, and b) reducing diagnosis and treatment delay. Several initial steps that public health officials could do in strengthening TB-PPM model in Indonesia could include: 1) identifying type(s) of private healthcare providers with TB-diagnostic and treatment capacities, 2) increasing TB treatment awareness among private sectors, 3) designing sustainable TB treatment training programs, 4) creating a linkage for private healthcare providers to the national TB notification or surveillance system, and 5) establishing a referral mechanism for complicated or severe TB cases from private to public sectors. However, despite these potentials, there are significant challenges related to sustainability. For instance, efforts to incorporating private sectors into TB care is complicated by Indonesia's funding gap, as highlighted by the WHO Global Tuberculosis Report 2020, which reported a gap of US$318 million for Indonesia's national strategic plans for TB [40]. High level commitment from the government is critical to overcome these challenges.

## Strengths and limitations

This study is subject to several limitations. First, the design of this study prevented us from assessing changes happening between the two studies' periods. Consequently, we might have missed HCFs and/or PPs that were not identified during INSTEP study but no longer operating during COVET study periods, and were not properly registered in the local health department. This prevented us from justifying whether changes observed from before vs. during the COVID-19 pandemic were due to the effect of COVID-19 pandemic itself, or the underlying growth trend of private sectors in Indonesia. Additionally, we conducted the survey during the earlier/mid-phase of the pandemic, preventing us from observing further changes after the COVID-19 vaccine was widely available and the pandemic situation was much better

controlled. Further studies are needed to understand whether the changes observed in the earlier/mid-phase of the pandemic are consistent towards the end of/post-pandemic in Indonesia or other settings with similar TB/COVID burden. Despite this limitation, our finding may provide a snapshot of challenges faced by HCFs during the early phase of another potential future pandemic. Second, our response rate among PPs was ~73%. However, nonresponse bias was not a concern as the characteristics of PPs interviewed during INSTEP and COVET studies are generally similar (**Table 1** and **S3 Table**). Third, we relied on enumerators' devices without proper device calibration to collect spatial data, which could result in a measurement bias. To minimize this systematic error, we thoroughly cleaned the spatial data and revisited the HCFs location when location points on the map seemed off. Fourth, we experienced several disruptions during the data collection process due to movement restrictions imposed by the Indonesian government. As a result, we experienced a high turnover of study staff and survey enumerators. Consequently, it is plausible to have differing data quality throughout our study period. However, the effect of this on our study findings is likely to be negligible since we carefully trained our new enumerators and other study staff. Fifth, we were not able to assess the direct impact of COVID-19 on the TB cascade of care among private healthcare providers since we did not collect information on number of individuals seeking care for TB disease (i.e., individuals testing for TB disease, receiving TB diagnosis, starting, and completing TB treatment). To note, private sectors are not part of the Indonesia's national TB treatment program. Thus, basic TB treatment information (including whether directly observed therapy [DOT] were enforced), contact tracing or other information that would have been reported as part of the TB surveillance by private sectors were not available for us. Studies that combine programmatic data and information collected from cohort studies which prospectively follow individuals seeking TB care among private healthcare facilities are still warranted to quantify the effect of COVID-19 pandemic on the TB cascade of care. Another problem is that a large proportion in high incidence countries of private providers still need to participate in collaboration or follow recommended TB management practices [41]. Furthermore, many private healthcare facilities in Indonesia likely underreport cases to the National Tuberculosis Program (NTP). This underreporting is a critical issue, as it hampers the NTP's efforts to reduce the number of missing TB cases in the country [30]. Only 2% of private healthcare facilities were covered by DOTS services, including diagnosis [42]. Sixth, this study was conducted in an urban setting with high TB and COVID-19 burdens in Indonesia. Thus, our finding may not be generalizable to other settings with different TB and COVID-19 burdens, as well as different healthcare systems. Despite study limitations, our study was first to compare the geographical spread of and services offered by private healthcare providers and explore how it may impact TB care in a high TB burden setting in the early phase of the COVID-19 pandemic.

## Conclusion

This study confirms that private healthcare providers were adversely impacted by the COVID-19 pandemic, but some showed increased capacities that could potentially contribute to TB care in Indonesia. Overall, our findings underscored the need to build a systematic system (i.e., registry) to track operating HCFs and practicing PPs as we observed high turnover between the two study periods. More importantly, from our survey, we identified opportunities to expand TB care to private sectors, especially with the increasing number of diagnostic (i.e., laboratory and radiology services) and other services (i.e., pharmacy and linkage to the universal health insurance) observed during the COVID-19 pandemic. Our findings provide a deeper understanding of the complexities involved in private sector engagement for TB care. Results of our study can be used to inform, rethink, and redesign private-sector engagement

strategies which aim to recover from the major disruption caused by the ongoing COVID-19 crisis in the context of TB care.

## Supporting information

**S1 Checklist. Inclusivity in global research.**
(DOCX)

**S1 Fig. Changes in services provided by private healthcare facilities interviewed during INSTEP and COVET studies stratified by types of healthcare facilities.**
(DOCX)

**S2 Fig. Qualifications and proportion of private practitioners in COVET study who managed patients with respiratory tract infection (RTI) symptoms included and those who encountered TB patients (N = 476).**
(DOCX)

**S1 Table. Characteristics of private practitioners who managed patients with RTI symptoms in INSTEP (2017) and COVET (2021) studies.**
(DOCX)

**S2 Table. (Extension of Table 2) Impact of COVID-19 pandemic on services offered at private healthcare facilities in Bandung stratified by types of healthcare providers during COVET (N = 235).**
(DOCX)

**S3 Table. Characteristics of practicing private practitioners in INSTEP and COVET studies.**
(DOCX)

## Acknowledgments

We would like to thank the field enumerators who assisted with the data collection process. We would also like to thank our data manager, Nopi Susilawati, for her assistance in curating and maintaining the REDCap Application/Web database. Furthermore, we would also like to express our gratitude to Bandung Municipal Health Office for their assistance during our study implementation.

## Author Contributions

**Conceptualization:** Charity Oga-Omenka, Bony Wiem Lestari, Madhukar Pai, Bachti Alisjahbana.

**Formal analysis:** Rodiah Widarna, Nur Afifah, Argita Dyah Salindri, Bony Wiem Lestari, Bachti Alisjahbana.

**Funding acquisition:** Madhukar Pai.

**Investigation:** Rodiah Widarna, Bachti Alisjahbana.

**Methodology:** Rodiah Widarna, Charity Oga-Omenka, Bony Wiem Lestari, Madhukar Pai, Bachti Alisjahbana.

**Project administration:** Rodiah Widarna, Hanif Ahmad Kautsar Djunaedy.

**Software:** Hanif Ahmad Kautsar Djunaedy.

**Supervision:** Rodiah Widarna, Charity Oga-Omenka, Argita Dyah Salindri, Bony Wiem Lestari, Bachti Alisjahbana.

**Visualization:** Rodiah Widarna, Hanif Ahmad Kautsar Djunaedy, Argita Dyah Salindri, Bachti Alisjahbana.

**Writing – original draft:** Rodiah Widarna, Argita Dyah Salindri.

**Writing – review & editing:** Rodiah Widarna, Nur Afifah, Hanif Ahmad Kautsar Djunaedy, Angelina Sassi, Nathaly Aguilera Vasquez, Charity Oga-Omenka, Argita Dyah Salindri, Bony Wiem Lestari, Madhukar Pai, Bachti Alisjahbana.

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
