## [Decision Letter · Decision Letter 0]

14 May 2024

PGPH-D-24-00569

The shifting landscape of private healthcare providers before and during the COVID-19 pandemic and the potential impacts on tuberculosis care

Dear Dr. Alisjahbana,

Thank you for submitting your manuscript to PLOS Global Public Health. After careful consideration, we feel that it has merit but does not fully meet PLOS Global Public Health’s publication criteria as it currently stands. Therefore, we invite you to submit a revised version of the manuscript that addresses the points raised during the review process.:

The article tries to explore the correlation between the COVID-19 pandemic and TB-related services, specifically looking at  the private sector's involvement during this crisis. Even before covid, in many countries for a successful TB program the involvement of private sector has been sought after.  So this is a good move to find out the impact of involvement of private sector to enhance the TB program in spite of the constraints due to the pandemicYou may please mention  about the limitations due to the constraints by pandemic. As suggested by the reviewers we may  not be able to compare the situation before and after Covid, as by definition we have still not  reached a stage of after covid as new strains of the virus are still coming up. This part could be attempted later on when the pandemic might come to a complete stopPlease take care of all the minor revision points suggested by the reviewers. The major comments suggested may totally change your study aspects and the conclusions. You may keep in mind the major comments and can be utilised if you are planning a more elaborate extension of the study

We look forward to receiving your revised manuscript.

Kind regards,

Suma Krishnasastry, MBBS, MD,DNB

Academic Editor

Journal Requirements:

Additional Editor Comments (if provided):

Reviewers' comments:

Reviewer's Responses to Questions

**Comments to the Author**

1. Does this manuscript meet PLOS Global Public Health’s publication criteria? Is the manuscript technically sound, and do the data support the conclusions? The manuscript must describe methodologically and ethically rigorous research with conclusions that are appropriately drawn based on the data presented.

Reviewer #1: Yes

Reviewer #2: Yes

Reviewer #3: Yes

Reviewer #4: Yes

2. Has the statistical analysis been performed appropriately and rigorously?

Reviewer #1: Yes

Reviewer #2: Yes

Reviewer #3: Yes

Reviewer #4: Yes

3. Have the authors made all data underlying the findings in their manuscript fully available (please refer to the Data Availability Statement at the start of the manuscript PDF file)?

Reviewer #1: Yes

Reviewer #2: Yes

Reviewer #3: Yes

Reviewer #4: Yes

4. Is the manuscript presented in an intelligible fashion and written in standard English?

Reviewer #1: Yes

Reviewer #2: Yes

Reviewer #3: Yes

Reviewer #4: Yes

5. Review Comments to the Author

Reviewer #1: I would like to thank the authors for the effort they put into conducting this valuable study. Although it faced several limitations, it is valuable to many readers. Here are a few comments that came to my mind.

Would you please elaborate more on the questionnaire you used to conduct this study? Was it a validated questionnaire?

Also, as your study only includes data from the pre-COVID and COVID periods, I suggest adding a section to the discussion if there is any available data for the post-COVID era, either nationally or internationally.

Reviewer #2: Overall:

• The title and objectives mention "impact on TB," but the results don't fully support this claim. To assess the impact on TB services, the study design could have enrolled and study the impact amomng patients receiving TB treatment (DOTS centers) or specialists seeing potential TB patients. Using RTI symptoms as a proxy for TB symptoms during COVID-19 (which also has respiratory symptoms) might not be the most accurate approach. It is suggested consider revising the title and objectives to better reflect the actual findings, and if possible remove the word Impact on TB.

Specific Points:

1. Objectives: There's a discrepancy between the objectives in the abstract and the manuscript. Suggest revising the abstract to remove "impact" and reflect the actual objectives outlined in the manuscript.

“Abstract: to compare the spatial distribution of private providers and describe changes in characteristics and services offered during and before the COVID-19 pandemic, and explore its potential impact on tuberculosis care.”

“Manuscript: to describe a) the spatial distribution and the density of private healthcare facilities (HCFs) before and during the COVID-19 pandemic, b) changes in services offered by private HCFs and private practitioners (PPs) before and during the COVID-19 pandemic, and c) TB and COVID-19 related services offered by the private sectors during the COVID-19 pandemic in Bandung, Indonesia.”

2. Line 50: Consider using Cross-Sectional Study with Historical Comparison for better clarity.

3. Line 59: Specify the study period when mentioning the number of surveyed HCFs. (INSTEP or COVET)

4. Introduction:

o Introduce health system in Indonesia and importance of private healthcare facilities their services before and after COVID-19.

o Describe the COVID-19 restrictions (lockdowns, travel limitations, commerce etc.) that likely impacted healthcare delivery. My suggestion is to limit TB and its burden.

Methods:

5. Consider dividing the study duration into two distinct phases: 2017 INSTEP and COVET April 5th, 2021 - December 27th,2021.

6. Line 173-174: The inclusion of "TB-related services offered by private HCFs before and during the COVID- 19 pandemic. " as a primary outcome needs justification since the results might not fully address this.

7. Line 175: Clarify if the population size used for calculating HCF density remained the same in both phases (corresponding to CHC coverage areas) or if it reflects the actual population of CHCs during each phase.

8. Line 198: Specify whether the p-value should be a≤0.05 or p ≤0.05.

Results:

9. Line 208: ‘During the study period….’. Specify whether the results pertain to the INSTEP or COVET phase.

10. Line 284: Use "relative decrease" (7%) for better understanding. Since it's significant, consider "significantly decreased" instead of "slightly decreased."

11. Line 290 - 293: relative change is difficult to apprehend without observing table 1 footnote. Rephrase these lines for better clarity or reference the relevant table footnote for easier comprehension.

12. In line 284- Ensure consistency in how changes are expressed throughout the manuscript (absolute numbers vs. percentages).

13. The title, objectives, and some discussion points mention the "impact on TB," but the results lack strong evidence to support this claim. Consider revising these sections accordingly. Considering RTI symptoms as a proxy for TB symptoms is not appropriate, specially during COIVD-19 pandemic.

14. Table 1 has variables, Encountered TB patients. Mare increase in encountered TB patients statistics doesn’t mean managing TB patients. Table 2 has change in TB diagnostics services offered between tow period, which is not significant. Table 4 and Fig 5 have some statistics related to TB services during COVID-19 pandemic only. They don't offer sufficient information to estimate the impact on TB care.

15. Line 402: The data in Table 1 doesn't necessarily support the claim of a "substantial increase" in managing TB patients. Revise this section based on the actual data presented.

16. Line 416: ‘Our study findings also suggested that private sectors are constantly growing, and we observed increases in TB diagnostic and other services compared to our 2017 survey.’ Table 2 doesn't show a change in TB screening and testing services. Revise this section to accurately reflect the findings.

17. Sustainability: The discussion could acknowledge the challenges of incorporating the private sector into TB care given Indonesia's funding gap (WHO Global Tuberculosis Report 2020 observed gap of US$405 million for TB).

Reviewer #3: Overall: The research objectives are well-defined and align with the scope of the study. The article effectively communicates its research questions, providing information about the study objective (role of the health sector and impact of COVID-19 on TB services). Data Analysis: The data analysis addresses the research questions. The statistical methods employed are appropriate for the research design, contributing to the validity of the results. Both significant and some insignificant study-related findings are adequately explained within the text. Graphics and maps were well-presented and helpful to understand the context.

Some recommendations:

1. The article offers a comprehensive exploration of the correlation between the COVID-19 pandemic and TB-related services, shedding light on the private sector's involvement during this crisis. While the results appear to fulfill the study's objectives, it's important to note that factors such as the operational capacity of health facilities, the availability of medical personnel, medication supply chains, and overall healthcare services, all have been significantly affected by COVID-19. Thus, the inclusion of TB-control-specific measures and other TB-oriented services such as DOT, contact tracing, and TB surveillance is particularly noteworthy. If such services are not covered by private health facilities or such data were not available, it might be advisable to include in the limitation section.

2. In the setting section, the term "community health centers," known as Puskesmas, typically refers to public health facilities. However, this study's data were sourced from the COVET study, which focuses on the private health sector. It's possible that the study examined data from private health facilities within the catchment area of community health centers. Clarification regarding the definition of "community health areas" and "centers" within the context of this study would enhance clarity, especially for readers unfamiliar with the terminology.

3.Concerns about whether the data were open resource, publicly published, or approved for reuse by the owner should be stated.

4. In the Implications and Recommendations section:

The discussion section provides valuable insights into the integration of the private sector into TB control programs. Additionally, recommendations regarding the impact of COVID-19 and strategies for better preparedness in future outbreaks would be beneficial to include.

Reviewer #4: Hello authors

Congratulations upon this manuscript

Below are some areas that I believe need improvement

Introduction: it focuses a lot on TB care instead of the private facilities that the title generalizes on. Either make the title more TB forward or change about 2 paragraphs to focus on private facilities and their turnover rates.

Methodology; are all the hcfs within the chc areas all private? Or the total numbers categorize both private and government owned. This needs to be clarified since the study is focusing on private facilities

Please describe how your study assessed the PPs who were not recorded in the INSTEP study and are nolonger working especially those who may have worked in the facilities that are permanently closed

If the sampling for interviews for the PPs was purposive, wouldn't that bias the data if you're using it to form statistical comparisons with the INSTEP survey

Discussion

The first paragraph undermines the study by stating that no significant findings were made. I believe you could start with what significant findings were found then comparing it to the previous studies

Limitations

Is the study finding effect of number health facilities on TB care or effect of COVID on TB care. Please discuss that clearly .

6. PLOS authors have the option to publish the peer review history of their article (what does this mean?). If published, this will include your full peer review and any attached files.

**Do you want your identity to be public for this peer review?** For information about this choice, including consent withdrawal, please see our Privacy Policy.

Reviewer #1: No

Reviewer #2: No

Reviewer #3: **Yes: **Marwah Al-Zumair

Reviewer #4: No

---

## [Editor Report · Decision Letter 1]

27 Aug 2024

The shifting landscape of private healthcare providers before and during the COVID-19 pandemic: lessons to strengthen the private sectors engagement for future pandemic and tuberculosis care

PGPH-D-24-00569R1

Dear Dr. Alisjahbana,

We are pleased to inform you that your manuscript 'The shifting landscape of private healthcare providers before and during the COVID-19 pandemic: lessons to strengthen the private sectors engagement for future pandemic and tuberculosis care' has been provisionally accepted for publication in PLOS Global Public Health.

Best regards,

Suma Krishnasastry, MBBS, MD,DNB, FRCP (Edin)

Academic Editor